# Mirroring Perinatal Outcomes in a Romanian Adolescent Cohort of Pregnant Women from 2015 to 2021

**DOI:** 10.3390/diagnostics13132186

**Published:** 2023-06-27

**Authors:** Daniela Roxana Matasariu, Irina Dumitrascu, Iuliana Elena Bujor, Alexandra Elena Cristofor, Lucian Vasile Boiculese, Cristina Elena Mandici, Mihaela Grigore, Demetra Socolov, Florin Nechifor, Alexandra Ursache

**Affiliations:** 1Department of Obstetrics and Gynecology, University of Medicine and Pharmacy “Grigore T. Popa”, 700115 Iasi, Romania; daniela.matasariu@umfiasi.ro (D.R.M.); irina.dumitrascu@umfiasi.ro (I.D.); alexandra-elena_s_mihaila@d.umfiasi.ro (A.E.C.); cristina-elena_i_tanasa@d.umfiasi.ro (C.E.M.); mihaela.grigore@umfiasi.ro (M.G.); demetra.socolov@umfiasi.ro (D.S.); alexandra.ursache@umfiasi.ro (A.U.); 2Department of Obstetrics and Gynecology, Cuza Vodă Hospital, 700038 Iasi, Romania; 3Biostatistics, Department of Preventive Medicine and Interdisciplinarity, University of Medicine and Pharmacy “Grigore T. Popa”, 700115 Iasi, Romania; lboiculese@gmail.com; 4Department Clinics, Faculty of Veterinary Medicine, Iasi University of Life Sciences (IULS), 700489 Iasi, Romania

**Keywords:** adolescent pregnancy, hypertension, postpartum hemorrhage, preterm labor, diabetes, cesarean section, perineal laceration, neonatal intensive care unit, pregnancy complications

## Abstract

Although the rates of adolescent pregnancies appear to have dropped according to the World Health Organization (WHO), the decrease in the age of the first menarche and better nutrition seems to contribute to the otherwise high rate of adolescent pregnancy worldwide, despite the efforts of different organizations to improve upon this trend. We conducted a population-based retrospective cohort study from January 2015 to December 2021 using our hospitals’ database. We totaled 2.954 adolescent and 6.802 adult pregnancies. First, we compared younger adolescents’ outcomes with those of older adolescents, as well as with adolescents aged between 18 and 19 years old; secondly, we compared adolescent pregnancies with adult ones. We detected higher percentages of cephalo-pelvic disproportion (43.2%), cervical dystocia (20.7%), and twin pregnancy (2.7%) in underage adolescents compared with 32%, 14.1%, and 1% in older underage adolescents, respectively, and 15.3%, 3.1%, and 0.6% in older ones. As teens became older, the likelihood of malpresentations and previous C-sections rose, whereas the likelihood of vaginal lacerations declined. When comparing adolescents with adult women, we found more cases that required episiotomy (48.1% compared with 34.6%), instrumental delivery (2.1% compared with 1%), and cervical laceration (10.7% compared with 8.4%) in the adolescent group, but the rates of malpresentation (11.4% compared with 13.5%), previous C-section (13.9% compared with 17.7%), and placenta and vasa praevia (4.5.6% compared with 14%) were higher in the adult women group. Adolescent pregnancy is prone to being associated with higher risks and complications and continues to represent a challenge for our medical system.

## 1. Introduction

An adolescent is considered to be a young person between 10 and 19 years of age in a transition between an immature condition and a mature one, which thus prepares them for reproduction. Adolescents can be divided into younger adolescents, between 10 and 14 years old, and older adolescents, between 15 and 19 years old. In Romania, a teenager is defined as a girl under 18 years old. To manage to obtain comparable results to those mentioned in the literature, we used the age of 19 years old stated in all the other studies as a cut-off. Even in developed countries, not to mention in low-income or middle-income ones, adolescent pregnancy remains an old but still present burden for the healthcare system. In total, 11% of all global births involve adolescents, with the vast majority of them taking place in low- and middle-income countries [1].

According to the WHO’s (World Health Organization’s) latest data, Algeria has the lowest rate of adolescent births (11.4 births per 1000 women in the corresponding age group) and Niger has the highest (165.37). The United States and Canada have the lowest rates of adolescent pregnancies in North America (14.4 births per 1000 women in the respective age group in the United States and 6.31 births in Canada), with Nicaragua having the highest in South America (82.47). Somalia occupies the highest position among the nations in the eastern Mediterranean, with 114.19 births per 1000 women in the corresponding age group, while the United Arab Emirates is the lowest, with 2.77. In South-East Asia, Bangladesh has the highest rate of adolescent pregnancies, with 71.75 births per 1000 women in that age group, and the Democratic People’s Republic of Korea has the lowest rate, with 2.59 births. In terms of Europe, Tajikistan and Azerbaijan take the top two positions, with 44.29 and 40.20 births per 1000 women in the corresponding age group. Bulgaria and Romania follow closely behind, with 37.80 and 34.79 births per 1000 women in the corresponding age group. When analyzing only EU countries, however, Romania ranks second. Denmark has the lowest rate of teenage pregnancies in Europe, with 1.73 births per 1000 women in the relevant age group [2].

In 2021, there were 34.7 births per 1000 women in the 15-to-19-year-old age group and 1.4 births per 1000 women in the 10-to-14-year-old age group, according to our country’s statistics. In Europe, Romania has the highest prevalence of adolescent pregnancies among teenagers between the ages of 10 and 14. In our nation, the overall rate of adolescent pregnancies dropped from 11.79% in 2009 to 9.48% in 2018, reaching 9.15% in 2021 [3,4]. In a large number of industrialized and developing nations, pregnancy becomes a source of increased morbidity during the gestational stage, as well as after delivery and the postpartum phase. It continues to be the leading cause of death for teenagers worldwide between the ages of 15 and 19, with a risk that is five times higher for those under the age of 15 [5]. The social, economic, and educational ramifications of an intended or unexpected pregnancy in this vulnerable age group are additional factors that we need to take into account when attempting to assess the worldwide effects. Teenagers appear to be more prone to brief recurrent pregnancies, especially in the first two years following their first delivery [6]. According to statistics, up to 70,000 adolescent deaths related to childbirth and pregnancies occur each year, with the majority occurring in middle-income nations [7].

The literature links adolescent pregnancies to several unfavorable maternal, perinatal, and neonatal outcomes [8]. Preterm delivery is becoming more common in this age group, according to numerous studies in the literature, and is often explained by insufficient antenatal care [6,7,8,9,10]. Studies also report high rates of PROM/PPROM (premature rupture of membranes/preterm premature rupture of membranes), anemia, maternal near-miss mortality (MNM), and maternal deaths in this age group [10,11,12,13,14]. There are many unfavorable prenatal, intrapartum, and postpartum events that have been connected to increased newborn morbidity and mortality. However, because of the elevated risk of obstructed labor, instrumental delivery, tissue trauma (vaginal, cervical, and perineal lacerations during labor), and delivery complications, there is evidence connecting adolescent births to an increase in obstetric intervention [13,14,15]. Among the unfavorable neonatal outcomes in adolescent pregnancies are low Apgar scores, respiratory distress, NICU admission, and delivery trauma. Prematurity continues to be the leading cause of newborn morbidity and mortality, placing a heavy financial burden on the healthcare system [9,10,11,12,13,14]. Another challenge that stands out and increases neonatal morbidity and mortality is impaired fetal growth, which is linked to low fetal weight in adolescent pregnancies [10,11,12,13]. Further research has shown that adolescent pregnancies typically have lower Apgar scores than adult pregnancies [13]. Due to the smaller fetal weight, adolescents deliver vaginally more commonly than adults, according to the majority of research [14]. In the literature, the causes of undesirable results were first solely attributed to the biological immaturity of teen mothers. Sample sizes, ethnic characteristics, and regional social and economic situations may be recognized as causes of the various unfavorable outcomes reported in studies [16]. Education, economic inequality, the availability of contraception, and sexual education all play a significant role in describing the variations between rural and urban locations with regard to adolescent pregnancy rates and outcomes [17]. The mounting data emphasize the critical role that social and economic factors play in the adverse outcomes found. The causes are still being debated, which adds to the already significant burden of low literacy in this subgroup and the insufficient addressability of the healthcare services [18].

According to a 2013 study by Part et al., the rate of adolescent pregnancy births in the pediatric population of Romania was 3.95% [19]. In 2021, the rate of adolescent pregnancy in our area ranged from 6.36% for older adolescents (between the ages of 15 and 19) to 0.11% in younger adolescents under the age of 15 [20]. Even though there have been and continue to be sustained efforts to improve outcomes and reduce the incidence of pregnancy in the pediatric population, despite the relatively low incidence, it continues to place a considerable burden on our healthcare system and is a significant source of high maternal–fetal morbidity and mortality. Because the literature shows abundant conflicting results regarding this specific topic, the need for precise, thoroughly researched data on this subject arises. Most complications occur in middle-income and low-income settings, motivating us to provide an accurate image of our region’s related particularities. In the study that follows, we tried to provide a picture of the difficulties related to childbirth occurring in adolescents in an obstetrics and gynecologist center of tertiary referral in northeastern Romania [10,19].

We further split adolescents into these three subgroups, because, as stated in the literature, younger adolescents and older adolescents show different risks and outcomes for both the mother and the fetus. In order to confirm or invalidate how each relates to our demographic, we decided to investigate each one separately [8,12].

## 2. Materials and Methods

### 2.1. Study Design

Our tertiary center hospital database was used to conduct a population-based retrospective cohort analysis from January 2015 to December 2021. A flowchart of the selection process for our adolescent cohort is shown in Figure 1. All of our institution’s data records pertaining to underage adolescent mothers and their infants were examined. The data were checked for validity or missing values and corrected by reviewing both the mothers’ and newborns’ medical records.

The gestational age of these pregnancies was determined by our hospital’s healthcare providers as the time between the delivery date and the first day of the patient’s last menstrual cycle, adjusted using either first- or second-trimester morphological scans or the patient’s sole ultrasound examination.


*Inclusion criteria*


We included in our study group all the adolescents between the ages of 12 and 19 who gave birth after 24 full weeks of gestation to a neonate weighing more than 500 g in our 7-year time-frame. In our hospital, no births from 10- or 11-year-old adolescents have been recorded. All the women who gave birth in the same time-frame aged between 20 and 24 were included in the control group.


*Exclusion criteria*


Due to age-related rising maternal–fetal complications and the higher frequency of assisted reproduction techniques, we did not include any pregnant women older than 24 in our control group.

We obtained approval from our hospital’s ethics committee to conduct this study (10,426/24 August 2021).

### 2.2. Variables of Interest

Maternal age was defined as the age of the mother in completed years at the time of delivery. We broadly categorized women into adults (20–24 years of age as a reference group) and adolescents (<19 years). The adolescents were further divided into three groups: younger underage adolescents (under 15 years of age), older underage adolescents (between 15 and 18 years old), and older adolescents (aged 18 and 19 years old).

Preterm birth is defined as birth that occurs between 24 and 36 weeks and 6 days of gestation. The WHO further divides preterm births into three categories: extremely preterm (less than 28 weeks); very preterm (28 to less than 32 weeks); and moderate-to-late preterm (32 to 37 weeks) [21].

We defined PROM as the rupture of the membranes prior to the beginning of labor and PPROM as membrane rupture occurring earlier than 37 weeks of gestation [22].

We also defined severe laceration at birth as a third- or fourth-degree laceration.

A low Apgar score was defined as a score of less than 7 at 5 min [23].

All newborn children that passed away within the first week following birth were referred to as neonatal deaths [24].

Anemia was defined as a hemoglobin value below 11 mg/dL [25].

### 2.3. The Ultrasound Diagnostic Criteria

-Fetus according to gestational age (AGA)—fetuses with a birthweight between the 10th and 90th percentile for the gestational age [26].-Small for gestational age (SGA)—fetuses with birthweight less than the 10th percentile for the gestational age, but higher than the 3rd percentile, in accordance with their genetic growth potential [27].-Fetal growth restriction (FGR) was classified as being:
-Moderate FGR—fetuses that fail to achieve their genetic inherited growth potential, with an estimated fetal weight between the 3rd and the 9th percentile compared with normal fetal weight for the gestational age [27];-Severe FGR—an estimated fetal weight at or below the 3rd percentile compared with normal fetal weight for the gestational age [27];-Large for gestational age (LGA)—birthweight >90th percentile for gestational age [28].

We examined the prenatal ultrasonography growth curve from the patients’ medical records to make a distinction between moderate FGA and SGA. To distinguish between both of these conditions in cases where there were no medical records because the patients were unable to reach healthcare professionals, we consulted with our neonatologists.

We evaluated the following maternal variables: gestational age at birth, the incidence of PROM/PPROM, the prevalence of anemia, and MNM. The Apgar score and admission to the neonatal intensive care unit (NICU) were the perinatal outcomes examined. The peripartum outcomes evaluated were: the prevalence of C-sections (cesarean sections) and their determinant cause, the requirement for performing an episiotomy, the need for an operative vaginal birth, and postpartum maternal complications.

Among the maternal outcome variables, the most important one remained MNM. The criteria for MNM were consistent with the WHO recommendations. MNM serves as a very helpful instrument to monitor and enhance maternal medical care, reflecting the quality of the healthcare system. We included here postpartum hemorrhage (soft birth canal lacerations, uterine atony, and retained placenta) with more than 500 mL blood loss during vaginal delivery or 1000 mL during C-section, the necessity of transfusion, and hemostatic hysterectomy. The women that survived these severe complications were considered to be near-miss cases. MNM was considered to be a better reflection of healthcare services [29,30].

### 2.4. Statistical Analysis

The SPSS application, version 24 (IBM Corp. Released 2016. IBM SPSS Statistics for Windows, Version 24.0. Armonk, NY, USA: IBM Corp.), was used for data analysis.

For each type of variable, separately, the statistics used to describe the data were absolute and relative frequencies or means and standard deviations. The frequencies of the categorical variables were compared, and the average was taken into consideration for the numerical data. In contingency tables, the Chi-square test was applied by default. Fisher’s exact test was used if the consistency criteria were not satisfied.

The ANOVA method for more than two samples (with post hoc tests if significance was reached) and t-test for two samples were used for continuous variables (ratio scale or interval). To check for homoscedasticity, the Levene test was therefore used to determine whether the variances were equal.

We evaluated the applied tests’ power. The G*Power version 3.1.9.6 application (free program, created by Franz Faul, Univ. Kiel, Germany 2020) completed the post hoc procedures for achieving power computations [31].

We discovered statistical power for t-tests between 0.06 and 0.46, which was less than 0.8, the conventional cutoff. We discovered power for Chi-square tests or Fisher’s exact tests between 0.05 and 0.91 limits.

## 3. Results

Striving to convey our findings in a methodical manner, we will divide them into maternal, perinatal, and neonatal outcomes. Firstly, we will compare the results of three groups of pregnant adolescent women: the first group included all cases found to be less than 15 years old, the second group included all cases found to be pregnant adolescents between the ages of 15 and 17, and the last group included all the cases aged between 18 and 19 years old. We chose to use this classification since many young women between the ages of 18 and 20 in our country give birth, the majority of them without any complications. Although the 18–19 age range in our country is no longer regarded to be part of adolescence, we included it in our analysis to provide similar and comparable data to those found in the literature. After the application of the inclusion and exclusion criteria, we registered 2.954 pregnant teenagers. A total of 6.802 pregnant adult women between the ages of 20 and 24 were identified. The flow-chart in Figure 1 presents our included cases per year of study.

As anticipated, the majority of our teenagers came from rural areas, with 73.5% being younger underage adolescents under 15 years of age, 80.8% being older underage adolescents between the ages of 16 and 17, and 86% being older adolescents between 18 and 19 years old (Table 1). With *p*-values of 0.01 and <0.001, respectively, these differences were statistically significant. Additionally, the majority of them, 97.5% from those under the age of 15, 88.7% from those between the ages of 15 and 18, and 34.5% from the last adolescent group, were primiparous (Table 1).

Although the younger groups, those under the age of 18, had statistically significantly more preterm births, our older adolescent group had a statistically significant increased number of PROM/PPROM cases than the younger two groups. When we examined antepartum hemoglobin and hematocrit levels, there were no statistically significant differences between our three adolescent groups, but the frequency of postpartum anemia was statistically significantly higher in our younger groups, with *p*-values of less than 0.01 regardless of birth type (Table 2).

As expected, the first adolescent group had a higher rate of cephalo-pelvic disproportion, 43.2%, with a *p*-value of less than 0.001, while the rate was 15.3% in the third adolescent group. Similar results were observed when analyzing cervical dystocia and twin pregnancy rates, both of which had higher values in younger adolescents. Placenta and vasa praevia incidence seemed to decrease with age when analyzing underage adolescents, but reached statistically significantly higher values in the third adolescent group (Table 3). We observed a higher incidence of malpresentation and previous C-section births (*p* < 0.001) in our third group of adolescent pregnancies compared with the first two. There were no statistically significant differences between our two groups of pregnant adolescents concerning the modality of birth, the need for instrumental delivery, episiotomy, instrumental curettage, or the incidence of perineal or cervical lacerations (Table 3). The situation was different when we evaluated the rates of vaginal lacerations. While there was a tendency for fewer vaginal lacerations in the third group compared with the second, there was still a statistically significant increase in incidence when compared with our younger underage adolescents (Table 3).

Regardless of age, more than 80% of our included adolescents delivered at term, with the incidence of prematurity decreasing from our first adolescent group to our third one (Table 4). The majority of infants were AGA when it came to fetal growth, but we found greater SGA occurrence in our younger underage adolescents. Underage adolescents exhibited statistically significantly higher rates of FGR, with our second group having the highest value. Our findings reveal a statistically significant ascending age trend in terms of LGA percentages. When comparing fetal presentation at birth and all three groups’ findings concerning both cephalic and breech presentations, we detected a peak of breech presentation incidence in our second adolescent group (Table 4). Regarding neonatal NICU admission and its causes (respiratory distress, low birth weight), or the Apgar score, there were no statistically significant differences between the three adolescent groups (Table 4).

Analyzing MNM occurrence, there were no statistically significant differences between our adolescent groups in terms of the need for hemostatic hysterectomy and blood transfusion. The maternal and neonatal mortality rates had such low values in our hospital that we could not perform any statistical analysis; we only present our data in Table 5. The adolescent pregnant groups registered a total of 19 antepartum fetal deaths, the number of deaths increasing from 2 in our first group and reaching 10 in our third, older adolescent one.

To summarize our pregnant adolescent results:-Our younger underage adolescent group had statistically significantly higher numbers of SGA fetuses, cephalo-pelvic disproportion, twin pregnancy, cervical dystocia, and fewer malpresentations;-Our second group of older underage adolescents had statistically significantly higher numbers of FGR cases, malpresentations, and vaginal lacerations;-Our third, older adolescent group had statistically significantly higher acute fetal distress, previous C-section, and placenta and vasa praevia cases compared with the first two adolescent groups, and more vaginal lacerations.

Then, we made a comparison between all of the pregnant adolescent participants in our study, divided into the three categories mentioned above, and pregnant adult women between the ages of 20 and 24. In terms of the patients’ area of origin, our data analysis did not reveal any statistically significant differences between the two groups.

The majority of newborns were AGA, while the adolescent population had statistically significantly more SGA and FGR instances. Between the two groups, the LGA percentage was remarkably similar (Table 6).

With a *p*-value of less than 0.01, the rates of C-section were statistically significantly lower in the adolescent group. When we looked at the reasons for C-sections, we found that there were statistically significantly more cases of acute fetal distress in the adult-group pregnancies. Teenagers had higher rates of cephalo-pelvic disproportion, cervical dystocia, and twin pregnancy compared with adult-group pregnancies, and lower rates of malpresentation, prior C-section, and placenta and vasa praevia (Table 7).

Between our two groups, there were statistically significant variations in the requirements for episiotomies and instrumental delivery. We noted more cases of instrumental delivery and episiotomy in the pregnant adolescent group, both with *p*-values less than 0.01. Adolescent pregnancies had a greater frequency of cervical laceration, with a *p*-value of less than 0.01. However, when we compared the frequencies of vaginal and perineal lacerations between the two groups, no statistically significant differences were found (Table 8).

With an incidence of 0.01%, the rates of hemostatic hysterectomy were comparable between the two groups. Similar to our adolescent pregnancies, we registered only a single postpartum neonatal fatality and 41 antepartum deaths in our adult pregnant women group.

To continue with our results:-The adolescent group had statistically significantly higher numbers of SGA and FGR fetuses than the pregnant adult women, lower rates of C-sections (with a higher incidence of cephalo-pelvic disproportion, cervical dystocia, and twin pregnancy), and higher rates of episiotomy and instrumental delivery;-The pregnant adults had statistically significantly higher cases of cervical lacerations and acute fetal distress, malpresentation, previous C-section, and placenta and vasa praevia.

## 4. Discussion

Our country’s rate of 34.7 adolescent pregnancies per 1000 women in the corresponding age group is comparable to that reported by WHO figures [2]. Despite a steady decline in overall adolescent pregnancy rates from 2009 to 2021, our nation continues to rank first in Europe for the highest incidence of adolescent pregnancy in the 10-to-14-year-old age group, which literature studies show has the highest maternal mortality, and the second after Bulgaria in terms of overall adolescent pregnancy rates [3,4]. The overall adolescent pregnancy rate in our north-eastern region was 6.47% in the pediatric population in 2021, about twice as high as the rate reported by Part et al. in 2013, and has been steadily declining since 2009 [20].

When we divided our teenage cohort into the WHO-defined age range, we noticed a 0.11% prevalence of adolescent pregnancy between 10 and 14 years old and a 6.36% rate between 15 and 19 years old. There were fewer pregnancies in the first age group than in the WHO statistics [2,3].

According to Dutta et al.’s study, the majority of the population of adolescents who underwent research came from rural areas. The percentage ranged from 73.5 to 86% of the patients who participated in our study, with age-related variations. Urban–rural discrepancies, however, diminished when adolescent pregnancies and adult pregnancies in those younger than 24 were compared (*p*-value = 0.461). This intriguing feature, which distinguishes our findings from those of other studies, may be explained by the fact that discrepancies between urban and rural areas are in some ways homogenized by national economic and literacy levels, access to healthcare facilities, and sexual education programs [29,32].

With 2.5% in our younger underage adolescent group, 11.4% in our older underage adolescents, and 65.48% in our older adolescent group, we found a higher percentage of multiparous women than Keskinoglu et al. did, totaling 44.82% from all underage pregnancies as opposed to 14.6% in their Turkish study. The lack of family planning services and sexual education was the cause of these recurrent adolescent pregnancies [33].


*Maternal outcomes:*


The adolescent premature birth incidence in our study was 12.8%, with 1.48% of adolescents demonstrating PROM/PPROM. In comparison to adult pregnancies, our adolescent group had a significantly greater rate of premature birth, particularly before 32 weeks of gestation, but a lower incidence of PROM/PPROM. PROM and preterm birth both have complex similar etiopathogenesis. Less resistance to clinical and subclinical infections will be present in a cervix that has not fully matured. With prematurely ruptured membranes resulting in premature birth, infection begins prostaglandin production [34]. Both premature birth and PROM are associated with a high frequency of inadequate prenatal care, delayed pregnancy recognition, and potential consequences. Our findings concur with those made public by Fleming et al., Kuma et al., and Jain et al. Premature birth is a significant financial burden on our healthcare system and is also linked to high morbidity and mortality rates [34,35,36].

The most frequently reported unfavorable outcome in the literature about pregnant adolescents is maternal anemia [37]. This condition proved statistically insignificant in our study despite being significantly more common in our group of pregnant adolescents than in pregnant adult women (*p*—0.222). According to certain research, the younger a teenage pregnant woman is, the lower her hemoglobin level will be because of the increased nutrition she needs to support the baby’s and young mother’s intense biological processes. Regardless of the birth type, we also found an indirect relationship between postpartum hemoglobin and hematocrit levels and teenage age, with more severe anemia cases in our underage adolescents (first and second groups) than in the third group. In both of our groups, anemia affected more than 50% of the pregnant women. Anemia is a primary cause of maternal mortality and unfavorable pregnancy outcomes. Premature birth, FGR, and even low Apgar scores are all caused by maternal anemia, which results from a combination of several causes, including infections, inadequate nutrition, and improper supplementing [37,38,39].


*Perinatal outcomes:*


The WHO reports that there are 13.4 million preterm births worldwide, with a prevalence of 4–16%. In 2009, 900,000 deaths were attributable to prematurity. The incidence of premature birth ranges from 5.4% in Lithuania and 5.6% in Sweden to 12% in Cyprus. In our country, there is an 8.4% estimated preterm birth rate [40]. We detected a 12.18% rate of premature births in our adolescent cohort and 2.63% in our adult cohort. It became clear that adolescent pregnancies have a significant contribution to our country’s premature birth rate. The average gestational age at delivery was similar between our adolescent and young women groups (38.04 weeks vs. 37.77 weeks).

The literature demonstrates controversial results related to the particularities of pregnancy evolution and birth in the adolescent population. Some studies report higher percentages of C-sections [41,42,43], some report similar C-section rates compared to adult pregnancies [44,45,46], and some studies report lower percentages of adolescent C-section birth [47,48]. As depicted in many previous studies, we observed a statistically significantly lower number of C-sections performed for various indications in the pregnant adolescent group when compared with pregnant women between 20 and 24 years old (*p*-value less than 0.01). The numbers are similar to those detected by Azevedo et al. in their study conducted in Sao Paulo, Brazil. The percentage of C-section adult births in our study was 36.95%, and that in our adolescent cohort it was 32.12%, by far exceeding the percentages observed by Katz Eriksen et al. in 2016 (14.3% C-section rate), Al-Haddabi et al. in 2014 (10% C-section rate), and Kawakita et al. in 2016 (14.97% C-section rate in younger adolescents and 20.72% in older adolescents) in their population studies [9,49,50,51,52].

Furthermore, due to the practice of defensive medicine in our nation, there was a high incidence of C-sections in patients of all age groups, leaving no clear, physiologically supported explanation for this observation. Despite the immaturity of their reproductive systems, the literature suggests that the lower C-section rates among adolescent pregnancies may be due to improved myometrial function and elasticity [44,53], increased connective tissue elasticity, decreased cervical compliance [44], lower fetal birth weight [47], physician preferences [33], and higher preterm delivery rates [54]. The lower incidence of C-sections among adolescents may be explained by the fact that the findings of our study are compatible with some of the above-mentioned aspects.

When examining the underlying causes for C-section separately, a decrease in cephalo-pelvic disproportion with patient age was observed, as was expected. Compared with adolescents, adult women experienced fewer of these instances. When compared with older adolescents, the same trend was noted among younger adolescents. There were differences in cervical dystocia as well, with the rate being two times higher in adolescent pregnancies than in adult ones. When examining cases of acute fetal distress, malpresentation, and twin pregnancy, we discovered similar results between adult and teenage pregnancies. Additionally, compared with the adult population, the adolescent group population was characterized by a higher incidence of placental and vasa praevia, possibly related to the subsequent high incidence of previous C-section births. As a result, the likelihood of cephalo-pelvic disproportion, one of the major causes of maternal morbidity and mortality, decreased with age, contrary to the findings of Ogawa et al. but similar to the results of Jain et al. [34,47].

Our three groups of pregnant adolescents did not differ significantly in terms of the need for an instrumental delivery, an episiotomy, an instrumental curettage, or the frequency of perineal or cervical lacerations, but the second and the third groups had statistically significantly more vaginal lacerations (*p*-value of 0.023). We identified statistically significant differences between teenage and adult birth complications in the need for an episiotomy and instrumental delivery. Although adult women had more cervical lacerations, adolescents were more likely to require an episiotomy, an instrumental delivery, or cervical laceration. The frequency of vaginal, perineal, and cervical lacerations in adolescent and adult females did not differ statistically significantly. Contrary to Nkwabong et al., we discovered statistically significantly higher rates of instrumental deliveries in adolescent women than in adult women (*p*-value less than 0.01), consistent with Derme et al., Kawakita et al., and Hoque et al. [46,52,55,56].

Nulliparity, operative delivery, birthweight over 3500 g, labor induction, and lengthier second-stage labor are some of the risk factors for perineal laceration [52,57]. This aspect is also marked by conflict in the literature. Our study’s findings on perineal lacerations, however, were similar to those of Njim et al. in their Cameroon study, as well as Fouelifack et al., who both reported similar percentages of perineal laceration when comparing adolescent births to adult ones, but in disagreement with Sánchez-vila et al., who found higher rates of perineal lacerations, and with Kawakita et al. As expected, adolescents were more likely to require an episiotomy and instrumental delivery, but they also had reduced incidence of perineal and vaginal lacerations, highlighting the protective function of an episiotomy even when instrumental delivery is performed. The statistically significantly greater incidence of cervical lacerations confirmed that the developing teenage cervix is more susceptible to cervical dystocia or laceration. Additionally, as they age, adolescents become more susceptible to vaginal lacerations, possibly because fetal birth weight increases and the need for instrumental delivery maintains itself [8,12,45,52,58].


*Neonatal outcomes:*


In comparison with adult pregnancies, we identified statistically significantly greater percentages of SGA and FGR in adolescent pregnant women (*p*-value less than 0.01), as well as similar proportions of LGA fetuses. Additionally, the first group of younger adolescents (17.2%) experienced more preterm births between 29 and 36 weeks of gestation than older adolescents (13.9% and 10.68%, respectively). In our adolescent group, the frequency of preterm birth decreased as age increased. The immaturity of the reproductive system is more likely to influence the type of birth in younger adolescents; nevertheless, as people get older, their reproductive systems gradually mature, which lessens its negative impacts. These aspects also emerged in our adolescent cohort, explaining the falling trend in C-section births [8,12].

We discovered no statistically significant differences between the younger and older adolescent groups when examining the immediate fetal outcome, taking into consideration the Apgar score, the necessity for NICU admission, and the reason why admission was necessary (respiratory distress, low birth weight). This conclusion emphasizes that poor fetal outcome is not directly associated with a younger age, as reported by Ogawa et al. Additionally, Moraes et al. in Zambia and Mubikayi et al. stated that birth asphyxia and lower Apgar scores are more common in newborn adolescents [47,59,60].


*Maternal Near-miss Mortality*


We discovered no statistically significant differences between our three adolescent groups when we examined antepartum hemoglobin and hematocrit levels, the requirement for blood transfusion, and hemostatic hysterectomy, which is consistent with Althabe et al.’s findings [8]. The only statistically significant difference noted was the higher incidence of postpartum anemia in younger underage adolescents compared with older ones, regardless of the birth type. The value of hemoglobin and hematocrit increased with adolescent age in our study. Although studies have linked pregnancy in this age group to higher rates of maternal morbidity and mortality, our data run counter to this. The results of our analysis of MNM between the two teenage groups did not reach statistical significance, contradicting the results of Çift et al. stating that problems are more likely to occur in very young mothers [61]. Although Keskinoglu et al. argue that adolescent pregnancy accounts for one-fourth of maternal mortality, they also found similar results when comparing pregnant adolescent deaths to other age groups. Nove et al. and Zang et al. did not find any records of adolescent maternal deaths in their studies [33,54,62].

There are limitations to the study we conducted. Without placing any emphasis on the socioeconomic factors and excluding the subjective factors and influences of family and community, we simply concentrated on disclosing significant medical features. Our research did not examine issues related to addictive habits like drinking, smoking, or drug usage. Future studies are required to accurately determine how these factors affect outcomes for the mother and the fetus.

## 5. Conclusions

Pregnancy among adolescents is frequently accompanied by greater risks and unfavorable outcomes. We attempted to reflect the unique characteristics of our country’s region with regard to teenage pregnancy and its potential complications.

There were particularities between younger and older adolescent pregnancies’ maternal and fetal outcomes. The younger adolescents had statistically significantly higher rates of SGA fetuses, cephalo-pelvic disproportion, twin pregnancies, and cervical dystocia compared with the older adolescents, who had statistically significant higher rates of FGR fetuses, malpresentation cases, placenta and vasa praevia, and vaginal lacerations.

We found higher numbers of SGA and FGR fetuses in adolescent pregnancies compared with adult pregnancies. We also found greater rates of episiotomy and instrumental delivery and cervical lacerations, and lower rates of C-sections (with a higher frequency of cephalo-pelvic disproportion, cervical dystocia, and twin pregnancy). The prevalence of severe fetal distress, malpresentation, prior C-sections, and placenta and vasa parevia was statistically significantly higher in our adult pregnant women group.

## Figures and Tables

**Figure 1 diagnostics-13-02186-f001:**
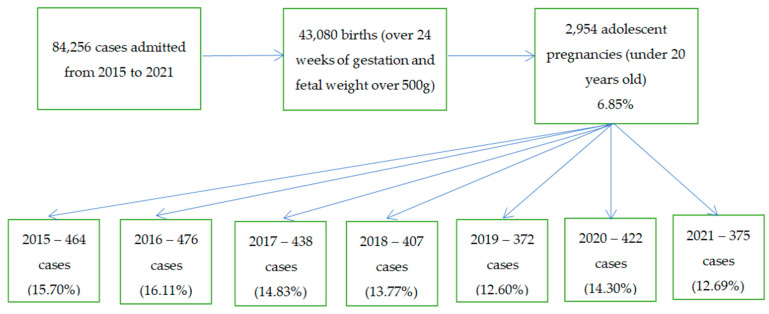
Flowchart of the cases we have included in our adolescent cohort.

**Table 1 diagnostics-13-02186-t001:** Maternal characteristics in our adolescent cohort.

	12 Years (*n* = 2)	13 Years (*n* = 12)	14 Years (*n* = 60)	15 Years (*n* = 205)	Total 1 (*n* = 279)	16 Years (*n* = 425)	17 Years (*n* = 378)	Total 2 (*n* = 803)	18 Years (*n* = 867)	19 Years (*n* = 1005)	Total 3 (*n* = 1872)	*p*-Value
Area												<0.01
rural		9 (75.0%)	38 (63.3%)	158 (77.1%)	205 (73.5%)	346 (81.4%)	303 (80.2%)	649 (80.8%)	756 (87%)	854 (85%)	1610 (86%)	P12 = 0.01
urban	2 (100.0%)	3 (25.0%)	22 (36.7%)	47 (22.9%)	74 (26.5%)	79 (18.6%)	75 (19.8%)	154 (19.2%)	111 (12.8%)	151 (15%)	262 (14%)	P13 ≤0.01 P23 = 0.01
Parity												<0.01
1	2 (100.0%)	12 (100.0%)	60 (100.0%)	198 (96.6%)	272 (97.5%)	387 (91.1%)	325 (86.0%)	712 (88.7%)	325 (37.48%)	321 (31.94%)	646 (34.50%)	P12 < 0.01
2				7 (3.4%)	7 (2.5%)	38 (8.9%)	50 (13.2%)	88 (11.0%)	475 (54.78%)	560 (55.72%)	1035 (55.28%)	P13 < 0.01
3							3 (0.8%)	3 (0.4%)	67 (7.72%)	124 (12.33%)	191 (10.20%)	P23 < 0.01

**Table 2 diagnostics-13-02186-t002:** Maternal outcomes in our adolescent cohort.

	12 Years (*n* = 2)	13 Years (*n* = 12)	14 Years (*n* = 60)	15 Years (*n* = 205)	Total 1 (*n* = 279)	16 Years (*n* = 425)	17 Years (*n* = 378)	Total 2 (*n* = 803)	18 Years (*n* = 867)	19 Years (*n* = 1005)	Total 3 (*n* = 1872)	*p*-Value
Preterm birth												<0.01
Yes		5 (41.7%)	9 (15%)	34 (16.6%)	48 (17.2%)	71 (16.7%)	41 (10.8%)	112 (13.9%)	101 (11.64%)	99 (9.85%)	200 (10.68%)	P12 = 0.187 **P13 < 0.01**
No	2 (100%)	7 (58.3%)	51 (85.0%)	171 (83.4%)	231 (82.8%)	354 (83.3%)	337 (89.2%)	691 (86.1%)	766 (88.35%)	906 (90.14%)	1672 (89.31%)	**P23 = 0.02**
PROM/PPROM												<0.01
Yes	0	0	1 (1.66%)	0	1 (0.4%)	4 (0.9%)	1 (0.3%)	5 (0.6%)	16 (1.84%)	22 (2.18)	38 (2.02%)	P12 = 0.61 P13 = 0.051
No	2	12	59 (98.33%)	205	278 (99.6%)	421 (99.1%)	377 (99.7%)	798 (99.4%)	851 (98.15%)	983 (97.81)	1834 (97.97%)	**P23 < 0.01**
Maternal anemia												
Hb antepartum (vaginal birth)	---	11.72 ± 0.54	11.65 ± 1.12	11.63 ± 1.05	11.64 ± 1.05	11.61 ± 1.21	11.79 ± 1.01	11.70 ± 1.12	11.72 ± 1.09	11.71 ± 1.15	11.72 ± 1.13	0.75 (A)
Hb postpartum (vaginal birth)	---	6.22 ± 5.72	8.84 ± 3.79	8.66 ± 3.93	8.63 ± 3.95	9.04 ± 4.16	9.41± 7.72	9.22 ± 6.14	10.51 ± 1.55	10.62 ± 1.44	10.57 ± 1.49	**P13 < 0.01**
*p*-value	---	0.082	**<0.001**	**<0.001**	**<0.001**	**<0.001**	**<0.001**	**<0.001**	**<0.001**	**<0.001**	**<0.001**	**P23 < 0.01**
Ht antepartum (vaginal birth)	---	34.38 ± 0.83	34.28 ± 4.67	34.92 ± 3.39	34.76 ± 3.65	34.68 ± 3.71	35.10 ± 3.45	34.90 ± 3.59	35.13 ± 3.25	35.09 ± 3.39	35.11 ± 3.33	0.31 (A)
Ht postpartum (vaginal birth)	---	18.06 ± 16.61	32.07 ± 34.34	25.57 ± 11.59	26.74 ± 19.09	26.36 ± 11.02	28.65 ± 28.54	27.47 ± 21.35	31.24 ± 4.98	31.49 ± 4.64	31.37 ± 4.80	**P13 < 0.01**
*p*-value	---	0.084	0.714	**<0.001**	**<0.001**	**<0.001**	**0.001**	**<0.001**	**<0.001**	**<0.001**	**<0.001**	**P23 < 0.01**
Hb antepartum (C-section)	10.95 ± 1.34	11.35 ± 0.89	11.56 ± 1.12	11.50 ± 1.04	11.48 ± 1.04	11.56 ± 1.37	11.61 ± 1.08	11.58 ± 1.25	11.66 ± 1.13	11.67 ± 1.18	11.66 ± 1.16	0.26 (A)
Hb postpartum (C-section)	9.30 ± 0.84	7.42 ± 5.16	8.49 ± 4.07	9.54 ± 3.07	9.17 ± 3.45	9.58 ± 3.28	10.32 ± 8.77	9.92 ± 6.39	10.70 ± 1.32	10.79 ± 1.27	10.75 ± 1.29	**P13 < 0.01**
*p*-value	0.133	0.095	**0.001**	**<0.001**	**<0.001**	**<0.001**	0.088	**<0.001**	**<0.001**	**<0.001**	**<0.001**	
Ht antepartum (C-section)	32.20 ± 2.26	33.21 ± 2.36	34.85 ± 3.65	34.48 ± 4.62	34.39 ± 3.10	34.48 ± 4.62	34.98 ± 3.16	34.73 ± 4.03	35.09 ± 3.24	34.93 ± 3.56	35.01 ± 3.42	0.16 (A)
Ht postpartum (C-section)	28.10 ± 0.98	28.10 ± 0.98	25.04 ± 12.08	28.14 ± 9.94	27.56 ± 11.26	28.14 ± 9.94	28.79 ± 9.42	28.50 ± 9.70	31.93 ± 4.17	32.01 ± 4.15	31.97 ± 4.16	**P13 < 0.01**
*p*-value	0.093	0.112	0.001	<0.001	**<0.001**	**<0.001**	**<0.001**	**<0.001**	**<0.001**	**<0.001**	**<0.001**	**P23 < 0.01**

*p*-values represent the results of the Chi-squared tests for frequency comparisons, or the results of the *t*-test (Student) or ANOVA (A) and post hoc tests for real continuous variables. The significant results are marked with bold.

**Table 3 diagnostics-13-02186-t003:** Perinatal outcomes in our adolescent cohort.

	12 Years (*n* = 2)	13 Years (*n* = 12)	14 Years (*n* = 60)	15 Years (*n* = 205)	Total 1 (*n* = 279)	16 Years (*n* = 425)	17 Years (*n* = 378)	Total 2 (*n* = 803)	18 Years (*n* = 867)	19 Years (*n* = 1005)	Total 3 (*n* = 1872)	*p*-Value
Presentation at birth												
Cephalic	2 (100%)	11 (91.7%)	59 (98.3%)	199 (97.1%)	271 (97.1%)	404 (95.1%)	357 (94.4%)	761 (94.8%)	844 (97.3%)	972 (96.7%)	1816 (97%)	**0.014**
Breech	0 (0%)	1 (8.3%)	1 (1.7%)	6 (2.9%)	8 (2.9%)	21 (4.9%)	21 (5.6%)	42 (5.25)	23 (2.7%)	33 (3.3%)	56 (3%)	
Types of delivery												
Vaginal	0 (0.0%)	5 (41.7%)	36 (60%)	127 (62%)	168 (60.2%)	257 (60.6%)	240 (63.5%)	497 (62%)	539 (62.2%)	633 (63%)	1172 (62.6%)	0.734
C-section	2 (100%)	7 (58.3%)	24 (40%)	78 (38%)	111 (39.8%)	167 (39.4%)	138 (36.5%)	305 (38%)	328 (37.8%)	372 (37%)	700 (37.4%)	
Indication for C-section												
acute fetal distress (abnormal fetal heart rate)	0 (0%)	0 (0%)	9 (37.5%)	11 (14.1%)	20 (18%)	35 (20.8%)	22 (15.9%)	57 (18.5%)	93 (28.4%)	93 (25%)	186 (26.6%)	**<0.01**
cephalo-pelvic disproportion	2 (100%)	2 (28.6%)	8 (33.3%)	36 (46.2%)	48 (43.2%)	53 (31.5%)	45 (32.6%)	98 (32%)	61 (18.6%)	46 (12.4%)	107 (15.3%)	
cervical dystocia	0 (0%)	4 (57.1%)	5 (20.8%)	14 (17.9%)	23 (20.7%)	26 (15.5%)	17 (12.3%)	43 (14.1%)	14 (4.3%)	8 (2.2%)	22 (3.1%)	
malpresentation	0 (0%)	1 (14.3%)	1 (4.2%)	4 (5.1%)	6 (5.4%)	18 (10.7%)	17 (12.3%)	35 (11.4%)	32 (9.8%)	54 (14.5%)	86 (12.3%)	
scarred uterus after C-section	0 (0%)	0 (0%)	0 (0%)	1 (1.3%)	1 (0.9%)	19 (11.3%)	23 (16.7%)	42 (13.7%)	49 (14.9%)	63 (16.9%)	112 (16%)	
placenta and vasa praevia	0 (0%)	0 (0%)	0 (0%)	2 (2.6%)	2 (1.8%)	2 (1.2%)	2 (1.4%)	4 (1.3%)	19 (5.8%)	28 (7.5%)	47 (6.7%)	
twin pregnancy	0 (0%)	0 (0%)	1 (4.2%)	2 (2.6%)	3 (2.7%)	3 (1.8%)	0 (0%)	3 (1%)	1 (0.3%)	3 (0.8%)	4 (0.6%)	
others	0 (0%)	0 (0%)	0 (0%)	8 (103%)	8 (7.2%)	12 (7.1%)	12 (8.7%)	24 (7.8%)	59 (18%)	77 (20.7%)	136 (19.4%)	
Instrumental delivery												
yes	0 (0%)	0 (0%)	2 (3.3%)	6 (2.9%)	8 (2.9%)	7 (1.6%)	15 (4%)	22 (2.7%)	12 (1.4%)	19 (1.9%)	31 (1.7%)	0.120
no	2 (100%)	12 (100%)	58 (96.7%)	199 (97.1%)	271 (97.1%)	418 (98.4%)	363 (96%)	781 (97.3%)	855 (98.6%)	986 (98.1%)	1841 (98.3%)	
Episiotomy												
yes	0 (0%)	3 (25%)	28 (46.7%)	103 (50.2%)	134 (48%)	203 (47.8%)	180 (47.6%)	383 (47.7%)	421 (48.6%)	482 (52%)	903 (48.2%)	0.967
no	2 (100%)	9 (75%)	32 (53.3%)	102 (49.8%)	145 (52%)	222 (52.2%)	198 (52.4%)	420 (52.3%)	446 (51.4%)	523 (52%)	969 (51.8%)	
Perineal laceration												
yes	0 (0%)	1 (8.3%)	5 (8.3%)	21 (10.2%)	27 (9.7%)	43 (10.1%)	48 (12.7%)	91 (11.3%)	117 (13.5%)	139 (13.8%)	256 (13.7%)	0.072
no	2 (100%)	11 (91.7%)	55 (91.7%)	184 (89.8%)	252 (90.3%)	382 (89.9%)	330 (87.3%)	712 (88.7%)	750 (86.5%)	866 (86.2%)	1616 (86.3%)	
Vaginal laceration												
yes	0 (0%)	0 (0%)	2 (3.3%)	4 (2%)	6 (2.2%)	29 (6.8%)	23 (6.1%)	52 (6.5%)	48 (5.5%)	55 (5.5%)	103 (5.5%)	**0.023**
no	2 (100%)	12 (100%)	58 (96.7%)	201 (98%)	273 (97.8%)	396 (93.2%)	355 (93.9%)	751 (93.5%)	819 (94.5%)	950 (94.5%)	1769 (94.5%)	
Cervical laceration												
yes	0 (0%)	0 (0%)	2 (3.3%)	17 (8.3%)	19 (6.8%)	44 (10.4%)	40 (10.6%)	84 (10.5%)	93 (10.7%)	119 (11.8%)	212 (11.3%)	0.073
no	2 (100%)	12 (100%)	58 (96.7%)	188 (91.7%)	260 (93.2%)	380 (89.6%)	336 (89.4%)	716 (89.5%)	774 (89.3%)	886 (88.2%)	1660 (88.7%)	
Instrumental curettage												
yes	0 (0%)	3 (25%)	10 (16.7%)	29 (14.1%)	42 (15.1%)	48 (11.3%)	51 (13.5%)	99 (12.4%)	99 (11.4%)	126 (12.5%)	225 (12%)	0.357
no	2 (100%)	9 (75%)	50 (83.3%)	176 (85.9%)	237 (84.9%)	376 (88.7%)	326 (86.5%)	702 (87.6%)	768 (88.6%)	879 (87.5%)	1647 (88%)	

*p*-values represent the results of the Chi-squared tests. Significance is marked with bold numbers.

**Table 4 diagnostics-13-02186-t004:** Neonatal outcomes in our adolescent cohort.

	12 Years (*n* = 2)	13 Years (*n* = 12)	14 Years (*n* = 60)	15 Years (*n* = 205)	Total 1 (*n* = 279)	16 Years (*n* = 425)	17 Years (*n* = 378)	Total 2 (*n* = 803)	18 Years (*n* = 867)	19 Years (*n* = 1005)	Total 3 (*n* = 1872)	*p*-Value
Birthweight (g)	3235 ± 7.071	2740 ± 565.615	2940.51 ± 551.501	3043.56 ± 545.493	3010.58 ± 548.276	3035.21 ± 577.081	3110.82 ± 521.444	3071.03 ± 552.377	3168.02± 583.36	3130.85± 567.32	3148.06± 574.95	**<0.01 (A)** P12 = 0.38 **P13 < 0.01** **P23 < 0.01**
AGA (according to gestational age)	2 (100.0%)	9 (75.0%)	44 (73.3%)	163 (79.5%)	218 (78.1%)	349 (82.1%)	303 (80.2%)	652 (81.2%)	752 (86.73%)	877 (87.26%)	1629 (87.01%)	**<0.01**
SGA (small for gestational age)		2 (16.7%)	7 (11.7%)	23 (11.2%)	32 (11.5%)	28 (6.6%)	28 (7.4%)	56 (7.0%)	33 (3.80%)	44 (4.37%)	74 (3.95%)	P12 = 0.12
FGR (fetal growth restriction)		1 (8.3%)	9 (15.0%)	13 (6.3%)	23 (8.2%)	37 (8.7%)	36 (9.5%)	73 (9.1%)	40 (4.61%)	50 (4.97%)	103 (5.50%)	**P13 < 0.01**
LGA (large for gestational age)				6 (2.9%)	6 (2.2%)	11 (2.6%)	11 (2.9%)	22 (2.7%)	32 (3.69%)	34 (3.38%)	66 (3.52%)	**P23 < 0.01**
(NICU) admission												
yes	0 (0%)	4 (33.3%)	11 (18.3%)	31 (15.1%)	46 (16.5)	69 (16.2%)	37 (9.8%)	106 (14%)	2 (0.23%)	1 (0.10%)	3 (0.16%)	0.173
no	2 (100%)	8 (66.7%)	49 (81.7%)	174 (84.9%)	233 (83.5)	356 (83.8%)	341 (90.2%)	930 (86%)	865 (99.76%)	1004 (99.90%)	1869 (99.83%)	
Reason for NICU admission												
neonatal respiratory distress	---	1 (25%)	3 (27.3%)	14 (45.2%)	18 (39.1%)	37 (53.6%)	18 (48.6%)	55 (51.9%)	1 (0.11%)		1 (0.05%)	0.148
low birthweight	---	3 (75%)	8 (72.7%)	17 (54.8%)	28 (60.9%)	32 (46.4%)	19 (51.4%)	51 (48.1%)	1 (0.11%)	1 (0.09%)	2 (0.1%)	
Apgar scores	8.50 ± 0.70	8 ± 2	8.08 ± 1.78	8.20 ± 1.31	8.17 ± 1.45	8.17 ± 1.33	8.23 ± 1.26	8.20 ± 1.29	8.27 ± 1.33	8.34 ± 1.22	8.31 ± 1.27	0.057 (A)

*p*-values represent the results of the Chi-squared test for frequency comparisons, or else the results of the ANOVA (A) or post hoc tests for real continuous variables. Significance is marked with bold numbers.

**Table 5 diagnostics-13-02186-t005:** Maternal near-miss factors in our adolescent cohort.

	12 Years (*n* = 2)	13 Years (*n* = 12)	14 Years (*n* = 60)	15 Years (*n* = 205)	Total 1 (*n* = 279)	16 Years (*n* = 425)	17 Years (*n* = 378)	Total 2 (*n* = 803)	18 Years (*n* = 867)	19 Years (*n* = 1005)	Total 3 (*n* = 1872)	*p*-Value
Mortality												
Postpartum neonatal mortality	0	0	0	0	0	0	0	0	0	1	0	-
Antepartum neonatal mortality	0	0	0	2 (0.97%)	2 (0.71%)	2 (0.47%)	4 (1.05%)	6 (0.74%)	6 (0.69%)	4 (0.39%)	10 (0.53%)	
Blood transfusion												0.499
yes	0 (0%)	0 (0%)	0 (0%)	5 (2.4%)	5 (1.8%)	6 (1.4%)	2 (0.5%)	8 (1%)	14 (1.6%)	14 (1.4%)	28 (1.5%)	
no	2 (100%)	12 (100%)	60 (100%)	200 (97.6%)	274 (98.2%)	419 (98.6%)	376 (99.5%)	795 (99%)	853 (98.4%)	991 (98.6%)	1844 (98.5%)	
Hemostatic hysterectomy												0.730
yes	0 (0%)	0 (0%)	0 (0%)	0 (0%)	0 (0%)	1 (0.2%)	0 (0%)	1 (0.1%)	1 (0.1%)	0 (0%)	1 (0.1%)	
no	2 (100%)	12 (100%)	60 (100%)	205 (100%)	279 (100%)	424 (99.8%)	378 (100%)	802 (99.9%)	866 (99.9%)	1005 (100%)	1871 (99.9%)	

*p*-values represent the results of the Chi-squared test or Fisher’s exact test (F) for frequency comparisons, or else the results of the *t*-test (Student) for real continuous variables.

**Table 6 diagnostics-13-02186-t006:** Neonatal outcomes.

Perinatal Outcomes	Adolescents (Frequency)	Women 20–24 Years Old (Frequency)	*p*-Value
Premature birth			
Yes	360 (12.18%)	179 (2.63%)	<0.01
No	2594 (87.81%)	6623 (97.36%)	
PROM/PPROM			
Yes	44 (1.48%)	326 (4.8%)	<0.01
No	2910 (98.51%)	6476 (95.20%)	
Birth weight (g)			
AGA (according to gestational age)	2501 (84.7%)	5942 (87.5%)	<0.01
SGA	160 (5.48%)	257 (3.77%)	
FGR	199 (6.73)	352 (5.17%)	
LGA (large for gestational age)	94 (3.2%)	238 (3.5%)	

**Table 7 diagnostics-13-02186-t007:** C-section indications.

	Adolescents (Frequency)	Women 20–24 Years Old (Frequency)	*p*-Value
Indication for C-section			
acute fetal distress (abnormal fetal heart rate)	263 (23.5%)	584 (19.1%)	<0.01
cephalo-pelvic disproportion	253 (22.6%)	373 (12.2%)	
cervical dystocia	88 (7.9%)	107 (3.5%)	
malpresentation	127 (11.4%)	412 (13.5%)	
scarred uterus after C-section	155 (13.9%)	541 (17.7%)	
placenta and vasa praevia	53 (4.7%)	477 (15.6%)	
twin pregnancy	10 (0.9%)	20 (0.7%)	
others	169 (15%)	537 (17.6%)	

**Table 8 diagnostics-13-02186-t008:** Perinatal complications.

	Adolescents (Frequency)	Young Women 18–24 Years Old (Frequency)	*p*-Value
Episiotomy			
yes	1420 (48.1%)	2352 (34.6%)	<0.01
no	1534 (51.9%)	4437 (65.4%)	
Perineal laceration			
yes	374 (12.7%)	910 (13.4%)	0.319
no	2580 (87.3%)	5879 (86.6%)	
Vaginal laceration			
yes	161 (5.5%)	394 (5.8%)	0.489
no	2793 (94.5%)	6395 (94.2%)	
Cervical laceration			
yes	315 (10.7%)	567 (8.4%)	<0.01
no	2636 (89.3%)	6222 (91.6%)	
Instrumental delivery/Operative vaginal delivery?			
yes	61 (2.1%)	69 (1.0%)	<0.01
no	2893 (97.9%)	6720 (99%)	

## Data Availability

The data used to support the findings of this study are available upon request.

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
