# Peer review of "Mirroring Perinatal Outcomes in a Romanian Adolescent Cohort of Pregnant Women from 2015 to 2021"

_diagnostics, 2023, doi:10.3390/diagnostics13132186_

Round 1

Reviewer 1 Report

Good manuscript on very important topic.

Would recommend clearly stating the primary outcome of interest and adding a power calculation to ensure your sample size is adequate to answer outcome of interest.

Would further elucidate rationale for young and old adolescent analysis.  

Line 43 confusing as it seems 18 is cutoff not 19

Line 48 - would state clearly WHO rates for adolescent pregnancy

Line 55 - unclear - grammatical issue?

Line 96 - is cost of adolescent pregnancy in Romania known?

Line 111 - any data on miscarriage and ectopic rates?

Discussion - would state how your rates compare to WHO rates as this is rationale for your study.

Minor issues with grammar and abbreviations (NICU, etc)

Reviewer 2 Report

Dear Authors, this is an interesting manuscript presenting outcomes of adolescent pregnant women divided into 2 subgroups (up to 15yo and 16-17yo) and compared with 1-24 young pregnant women in a large cohort of 9756 patients who delivered in the same Romanian tertiary hospital between 2015-2021. To improve your paper, the following suggestion might be helpful.   line 2-3: the title should be adjusted to your data e.g. Perinatal outcomes of Romanian adolescent pregnancies in 2015-2021. line 39: 1. Background - should be rather an Introduction and be more relevant to your study and factors you included in the analysis. Present more details what is known so far about adolescent deliveries in developed and developing countries as you studied mainly perinatal outcomes and we will know what point you started at. Clarify why you decided to check the perinatal outcomes in the two adolescent subgroups (<=15yo and 16-17yo), give the incidence of adolescent birth rate in Romania and/or Europe. line 43-44: what is the reason for the different definition, make efforts to adjust it to WHO definition in your study for comparison with other authors. line  48-82: to be removed as irrelevant to your subject. line 94-102: irrelevant to what you analyzed, make appropriate goals of your study.   line 104: make the comprehensive flowchart of your cases included in the study line 117-125: include to  Variables of interest and simply list the categories with reference line 129-136: should be in flowchart and Study design line 137-148: make 3 categories of your data to present in 3 separate tables: maternal characteristics, perinatal outcomes,  maternal near-miss factors line 149-162: rewrite the description of your statistical analysis in a more comprehensive and standardized way (see other published manuscripts for comparison).   line 164-167: use only present and past tenses accordingly, rewrite the description of your results. line 170: use p value in a standard way, 0.01 not 0.010. line 176: FGR is not the same as SGA, do not mix them, make the clear differentiation according to current definitions. line 199 table 1: divide into 3 tables (tab.1-3) as pointed above, remove  <= 15 years > 15 years from the top of columns as unnecessary, gestation age presented unclearly and numbers should be presented along with mathematical rules after dots, what is PROM/PPROM rate in your cohort, insert the data to the relevant table. line 213-214: Fig 2 change into table 4  for better visualisation so we could see proportions of AGA, LGA, SGA in 3 groups with p values. line 222-223: Fig 3 change into table 5 for better visualisation so we could see indications for CC in 3 groups with p values line 233-234: Fig 4 change into table 6 for better visualisation so we could see complications of VB in 3 groups with p values   line 236-263: Discussion is too long. You should focus on your results and compare them with others pointing similarities and differences, and trying to explain them rationally. Rewrite the chapter and remove philosophical aspects as you did not investigate them.   line 364: Conclusions are far from what you studied and found, focus on perinatal outcomes and MNM data to help clinicians take care of adolescent deliveries. Your data are very interesting and could be useful in daily clinical practice but you need to reveal them appropriately.   line 378: A.U. and F.N.; formal analysis - what does it mean? line 380: F.N.; funding acquisition and line 383  Funding: This research received no external funding - explain the funding and the role of FN author in the manuscript line 387:  Informed Consent Statement: Written informed consent has been obtained from the patients to publish this paper. - was it prospective or retrospective analysis? If the latter you do not need consents which are very hard to be signed by patients.

You may want to check your manuscript with recommended native editors.

Round 2

Reviewer 1 Report

There is a spelling error in the title now.  Overally comments and edits addressed concerns

see above

Author Response

Dear Reviewer, 

We want to start by sincerely thanking you for taking the time to assess our work and bring it up to the standards of such a prestigious journal.

The error in the title has been rectified.

Best whishes,

Elena Iuliana Bujor, MD
Department of Obstetrics and Gynecology
University of Medicine and Pharmacy "Grigore T. Popa"
16, University Street, Iasi, 700115, Romania
Phone: +40744.243.489
E-mail: bujor.iuliana@gmail.com

Reviewer 2 Report

Dear Authors,

  thank you for your significant efforts to make the manuscript eligible for publications in Diagnostics.   However, despite having very interesting data and results, still the manuscript is not written in a scientific way. It looks more like popular science information in a newspaper than a solid systematic clinical research presentation. General remarks: First, you must refer very carefully to Instructions for Authors on how to construct your paper. Second, you need to engage a specialist in obstetrics and gynecology or perinatologist to ensure your study group and results are described adequately in modern terminology and definitions. Third, the written English must be corrected by a native and/or professional writer/editor who is experienced in producing standardized research papers. Finally, reconsider statistical analysis according to the medical journal rules as you mixed several standards which makes the results difficult to understand and conclude. Always try to provide corrected versions with the option to hide the changes especially cosmetic ones.   Specific remarks: Abstract: is missing data which you found and calculated thus is not informative for users. Reconstruct it so a reader can benefit from just having time to look at your numerical results and conclusions.   Introduction: is still missing what is known in literature about your exact aims you want to focus on i.e maternal, perinatal and neonatal outcomes (MPN) in 2 adolescent subgroups. You should show what is stored so far in world libraries, present WHO data, European data, other continents data, your country data and  why you wanted to do the study in a specific group of pregnant women. Avoid philosophical divagations and newspaper style of thinking. Just focus on your goals: MPN outcomes in adolescent population in Romanian 3rd level hospital which could be or not representative for the whole country in comparison to other countries' data. Abandon the local Romanian definition of adolescent pregnancy and by using WHO definition include women 18yrs old in your study group as you have the database. Recalculation required. Explain the huge difference in adolescent pregnancies rate between WHO and Romania. Make clear the rationale why you wanted to write the paper.   Material and methods. Study design: lack of inclusion and exclusion criteria, what percentage of total population was included, flow chart unreadable due to poor sharpness, clarify if prospective or retrospective study. Cohort description: incomprehensive, describe properly, you cannot have written consent from all patients after several years of study design and approval, if consent form signed before ethical approval it is irrelevant and the committee can waive the written consents if data are anonymous in retrospective study. Variables of interest: very unclear and confusing, describe what you exactly studied and prove by relevant references that it is comparable with other sources, write the chapter according to scientific rules, see how some of your selected references were written as an example. Diagnostic criteria:  remarks and comments as above, stress the importance and value of maternal near-miss mortality (MNM) as a superior outcome compared to others, present how many maternal, fetal and neonatal deaths you had in your cohort. Statistical Analysis: unclear and confusing, write it according to medical data rules.   Results: do not discuss but present your data, tables overloaded and unsystematic, make more tables, present as separate tables or figures data for different MPN outcomes. They are very interesting but poorly displayed. Make the data comparable with other papers which use WHO criteria otherwise you miss significance and importance. Sharpness of pictures is insufficient. Over- and misuse of words 'totalizing, totalized'.   Discussion: rewrite according to the rules, compare your findings to others in a systematic manner as it should be listed in section Results to make it feasible for a reader to follow your idea.   Conclusions: are philosophical, not results-based and are lacking direct summary and recommendations for clinical readers, try to enhance the value and novelty of your findings in several points, show how adolescent pregnancy is safer or more dangerous for a woman, be objective not biased/prejudiced e.g. 1/ Adolescent women have higher incidence of SGA and FGR, 2/  There is a higher incidence of premature births in young adolescents (15.8%), compared to older ones (12.9%).   There is a valuable scientific potential in your raw data but it must be presented by a solid piece of paper in a recognizable structure. You may want to engage a professional editor available from the MDPI list investing your time and resources into substantial improvements of the manuscript. Alternatively the data can be published in your local national journal in native language.  

Must be improved and checked by professional editor e.g. from the MDPI list.

Author Response

Dear Reviewer,

We would first like to express our gratitude for your efforts to improve our work and make it suitable for publication in such a prestigious journal.

We hope the changes we made in response to your feedback live up to your expectations. We carefully followed all of your instructions when correcting our work. We sincerely appreciate you taking time out of your busy schedule to participate in the peer review process.

Best whishes,

Elena Iuliana Bujor, MD
Department of Obstetrics and Gynecology
University of Medicine and Pharmacy "Grigore T. Popa"
16, University Street, Iasi, 700115, Romania
Phone: +40744.243.489
E-mail: bujor.iuliana@gmail.com

Round 3

Reviewer 2 Report

Dear Authors,
your consistent efforts to improve the manuscript in my opinion made it eligible for publication.